# Effect of Teriparatide on Bone Mineral Density and Trabecular Bone Score in Type 2 Diabetic Patients with Osteoporosis: A Retrospective Cohort Study

**DOI:** 10.3390/medicina58040481

**Published:** 2022-03-26

**Authors:** Chihiro Munekawa, Yoshitaka Hashimoto, Noriyuki Kitagawa, Takafumi Osaka, Masahide Hamaguchi, Michiaki Fukui

**Affiliations:** 1Department of Endocrinology and Metabolism, Graduate School of Medical Science, Kyoto Prefectural University of Medicine, Kyoto 602-8566, Japan; c-mori@koto.kpu-m.ac.jp (C.M.); nori-kgw@koto.kpu-m.ac.jp (N.K.); tak-1314@koto.kpu-m.ac.jp (T.O.); mhama@koto.kpu-m.ac.jp (M.H.); michiaki@koto.kpu-m.ac.jp (M.F.); 2Department of Diabetology, Kameoka Municipal Hospital, Kameoka 621-8585, Japan; 3Department of Endocrinology and Diabetology, Ayabe City Hospital, Ayabe 623-0011, Japan

**Keywords:** teriparatide, bone mineral density, type 2 diabetes mellitus, bone quality, femoral neck

## Abstract

The BMDs of the lumbar spine, whole femur, and femoral neck and TBS were measured. Change in BMD or TBS was defined as the BMD or TBS at follow-up, performed 1 year after baseline, minus baseline BMD or TBS. *Results:* This retrospective cohort study included 93 patients, of whom 52 received no medication, 26 received bisphosphonates, and 15 received weekly teriparatide. BMD of the lumbar spine increased in all three groups. There was no change in BMD of the whole femur and femoral neck in the no medication and bisphosphonates groups, whereas the BMD of the whole femur (from 0.73 (0.15) to 0.74 (0.15) g/cm^2^, *p* = 0.011) and femoral neck (from 0.59 (0.16) to 0.60 (0.16) g/cm^2^, *p* = 0.011) in the teriparatide group increased. The change in BMD of the femoral neck (no medication; −0.002 (0.034) g/cm^2^, bisphosphonates; −0.0001 (0.024) g/cm^2^, and teriparatide; 0.017 (0.022) g/cm^2^, *p* = 0.091) or TBS (no medication; −0.007 (0.051), bisphosphonates; −0.058 (0.258), and teriparatide; 0.021 (0.044), *p* = 0.191) in the teriparatide group tended to be higher than that in the other groups, although there was no statistically significant difference. *Conclusions:* Teriparatide increased the BMD of the femoral neck and TBS in osteoporosis patients with type 2 diabetes mellitus, compared to bisphosphonates and no medication.

## 1. Introduction

Fractures are associated with a significant adverse effect on not only people’s quality of life, but also life expectancy [1,2]. Osteoporosis is an important risk factor for bone fractures and is one of the most important issues in preventive medicine today. In recent years, the risk of fractures in patients with type 2 diabetes mellitus (T2DM) has also gained increasing attention [3]. Fractures are more common among patients with diabetes than among those without; the relative risk of proximal femoral fractures is 1.3 to 2.8-fold higher in patients with T2DM [4,5,6,7].

Bone strength consists of bone mineral density (BMD) and bone quality [8,9]. Osteoporosis associated with T2DM is characterized by not only a decrease in BMD, but also a decrease in bone quality, which are thought to be responsible for the high risk of fracture in patients with diabetes [10,11]. Previous studies have shown that the decrease in bone quality is related to an increase in advanced glycation end-products (AGEs) due to hyperglycemia and deterioration of collagen cross-linking (increased pentosidine cross-linking) and reduced bone strength, independently of bone density [8,9]. Thus, there is a possibility that treatment focusing on factors related to bone quality may be more effective for osteoporosis patients with T2DM. 

Bisphosphonates are among the first-line treatments for osteoporosis and have been shown to improve BMD and reduce fracture risk [12]. Despite bisphosphonates’ well-known therapeutic potential, they have also displayed important side effects, among which is bisphosphonate-related osteonecrosis of the jaw, by targeting osteoclast activities as well as osteoblast and osteocyte behavior [13]. Recently, minimally invasive surgical treatment in early stages of medication-related osteonecrosis of the jaw has been recommended to prevent the evolution to more advanced stages and to promote downstaging of the lesions [14]. Conversely, teriparatide, also known as recombinant human parathyroid hormone (1–34), has anabolic properties and promotes bone formation [15]. It has been reported that teriparatide improves bone quality by inhibiting non-physiological cross-linking, such as pentosidine cross-linking caused by AGEs and other factors, and promoting physiological cross-linking through the production of osteoblast lysyl oxidase [16]. Furthermore, teriparatide has the effect of enhancing osteoblast function, inducing new bone matrix, and increasing BMD [17]. On the other hand, the most frequently reported adverse events were nausea (12.5%), arthralgia (11.7%), hypertension (8.9%), and headache (6.9%) [18]. Hypercalcemia was reported in 5% of the patients. The calcium concentration increases transiently, beginning 2 h after dosing, reaching a maximum concentration between 4 and 6 h (median increase, 0.4 mg/dL), decreasing 6 h after dosing, and returning to baseline values by 16 to 24 h after dosing. Persistent hypercalcemia was not observed in clinical trials of teriparatide. If persistent hypercalcemia is detected, teriparatide should be discontinued, and the patient should be assessed for the cause of the hypercalcemia [15].

A previous study reported that compared to patients without diabetes, T2DM patients treated with daily teriparatide treatment had a greater increase in the BMD of the femoral neck, and daily teriparatide reduced the incidence of new non-vertebral fractures [19].

However, the effect of weekly teriparatide on BMD has yet to be elucidated, especially compared to bisphosphonates in osteoporosis patients with T2DM. Therefore, this retrospective cohort study investigated the effect of teriparatide on BMD in patients with T2DM compared with bisphosphonates or no medication.

## 2. Materials and Methods

### 2.1. Study Population

This retrospective cohort study was the part of the KAMOGAWA-DM cohort study, an ongoing prospective cohort study [20]. All patients provided written informed consent. This study included T2DM patients with osteoporosis, confirmed by a dual-energy X-ray absorptiometry (DEXA) scan from November 2017 to August 2019. Patients were excluded if they did not have osteoporosis, were taking steroids or medications for osteoporosis, had rheumatoid arthritis or active malignancy, and if they had duplicate or inadequate data. None of the participants in this study had hyperthyroidism, Cushing syndrome, or hypogonadism. The local research ethics committee approved this study (No. RBMR-E-466-6 and ERB-C-774-2), and this study was carried out in accordance with the Declaration of Helsinki. 

### 2.2. Data Collection

Data of duration of diabetes, exercise habits, smoking status, and family history of diabetes were collected by a standardized questionnaire. According to the results of questionnaire, we divided the patients into smokers or non-smokers, and regular exercisers, who regularly played any type of sport at least once a week, or not. Medication data, including medications for diabetes and osteoporosis, were gathered from medical records. Further, venous blood was collected following an overnight fast, and the levels of uric acid, creatinine (Cr), high-density lipoprotein cholesterol, triglycerides, fasting plasma glucose, and hemoglobin A1c (HbA1c) were measured. Estimated glomerular filtration rate (eGFR) was calculated: eGFR = 194 × Cr^−1.094^ × age^−0.287^ (mL/min/1.73 m^2^) (×0.739, if female) [21]. 

### 2.3. Assessment of Bone Mineral Density, Trabecular Bone Score, Fractures, and Osteoporosis

BMD (g/cm^2^) was assessed at whole femur, femoral neck, and the lumbar spine (L2-L4) using DEXA (Hologic, Inc, Waltham, MA, USA). The percentage of young adult mean (YAM) and T-score, which is the number of standard deviations (SDs) between the mean BMD of the patient and the mean of the population compared to a gender- and race-matched reference population [22], were automatically analyzed. 

The trabecular bone score (TBS) of spine images (using the DEXA data of L1-L4) was assessed by the TBS Insight 2.2 software [23]. 

Osteoporosis was defined as (1) proximal femur fracture or vertebral fracture, (2) other fragility fracture and YAM ≤ 80%, and/or (3) YAM ≤ 70% and/or T-score ≤ −2.5 SD, according to the Japanese diagnostic criteria for primary osteoporosis [24] and the World Health Organization criteria [22]. Vertebral fractures were evaluated using spinal lateral radiographs according to the justification criteria for vertebral fractures [25]. 

### 2.4. Study Outcomes

In this study, patients with a diagnosis of osteoporosis were recommended treatment for osteoporosis, including bisphosphonates and teriparatide. Follow-up investigations were performed one year after diagnosis. The primary endpoint of this study was change in BMD of the lumbar spine, whole femur, and femoral neck. Secondary endpoints were change in HbA1c and change in TBS and the difference of change in BMD, change in BMD%, which were evaluated by change in BMD divided by BMD at baseline examination, HbA1c, or TBS among the groups.

### 2.5. Statistical Analysis

The normal distributions were evaluated by the Shapiro–Wilk test. We show the data as means (SD), median (1st quartile–3rd quartile), or frequencies of potential confounding variables. Patients were divided into the following three groups: (1) no medication, (2) bisphosphonates (alendronic acid or risedronate), and (3) teriparatide (Teribone™ 56.5 µg via subcutaneous injection once a week). Then, the differences in the continuous variables and categorical variables were evaluated by one-way ANOVA and Tukey–Kramer test or Kruskal–Wallis test and steel Dwass test, and the chi-square test, respectively. Differences between baseline and follow-up data were evaluated by the paired *t*-test. Data were also evaluated after excluding patients who stopped taking medications before the one-year follow-up. 

We used JMP version 13.2. software (SAS Institute Inc., Cary, NC, USA) for statistical analyses. Differences with *p* values < 0.05 were set as statistically significant.

## 3. Results

### 3.1. Study Participants

Finally, 93 patients, of whom 52 received no medication, 26 received bisphosphonates, and 15 received weekly teriparatide, were included in the analysis (Figure 1). In addition, among 28 patients who received bisphosphonates, 25 patients were able to continue for one year, and among the 21 patients who received weekly teriparatide, 10 patients were able to continue for one year.

### 3.2. Baseline Characteristics of Study Participants

Table 1 represents baseline characteristics of the study patients who received follow-up examinations. The mean (SD) age and duration of diabetes were 72.5 (7.6) and 12.9 (8.6) years, respectively, and 34.4% of the participants were men. The mean (SD) age of patients in the no medication, bisphosphonates, and teriparatide groups were 71.8 (7.1) years, 74.3 (7.7), and 71.9 (9.3) years, respectively (*p* = 0.378). HbA1c levels of patients in the no medication, bisphosphonates, and teriparatide groups were 7.4% (0.9), 7.4% (1.0), and 7.2% (1.1), respectively (*p* = 0.778).

### 3.3. Changes in BMD of the Lumbar Spine, Femoral Neck, and Whole Femur

Table 2 shows the values of BMD and TBS and their changes after the administration of no medication, bisphosphonates, and teriparatide. Although 93 patients were included in this study, the TBS data were available for 92 patients; thus, 92 patients were used for the analysis of TBS. BMD in the lumbar spine increased in the no medication (from 1.02 (0.19) to 1.04 (0.19) g/cm^2^, *p* = 0.013), bisphosphonates (from 0.83 (0.24) to 0.87 (0.25) g/cm^2^, *p* < 0.001), and teriparatide (from 0.86 (0.18) to 0.89 (0.16) g/cm^2^, *p* = 0.002) groups. 

There was no change in BMD of the whole femur and femoral neck in the no medication and bisphosphonates groups, whereas there was an increase in BMD of the whole femur (from 0.73 (0.15) to 0.74 (0.15) g/cm^2^, *p* = 0.011) and femoral neck (from 0.59 (0.16) to 0.60 (0.16) g/cm^2^, *p* = 0.011) in the teriparatide group. 

### 3.4. Change in HbA1c and TBS

Furthermore, TBS in the teriparatide group tended to increase (from 1.28 (0.08) to 1.30 (0.07), *p* = 0.097), although it did not reach statistical significance. 

The changes in HbA1c in the no medication, bisphosphonates, and teriparatide groups were 7.4 (0.9) to 7.6 (1.1)% (*p* = 0.012), 7.4 (1.0) to 7.7 (1.0)% (*p* = 0.056), and 7.3 (1.0) to 7.3 (0.9)% (*p* = 0.733), respectively.

### 3.5. Differences in Change in BMD of the Lumbar Spine, Femoral Neck, and Whole Femur, HbA1c and TBS among the Groups

There was a significant difference between the no medication group and bisphosphonate group in the change in BMD of the lumbar spine (0.011 (0.031) vs. 0.044 (0.044) g/cm^2^, *p* = 0.001, 1.102(3.155) vs. 5.519 (5.631)%, *p* = 0.001). Additionally, although there was no statistically significant difference, the change in BMD of the femoral neck in the teriparatide group tended to be higher than that in the other groups (no medication: −0.002 (0.034) g/cm^2^, −0.114 (5.315)%; bisphosphonates: −0.0001 (0.024) g/cm^2^, −0.216(4.751)%; teriparatide: 0.017 (0.022) g/cm^2^, 3.230(4.221)%; *p* = 0.091, *p* = 0.075).

Similarly, the change in TBS in the teriparatide group tended to be higher than in the other groups (no medication: −0.007 (0.051); bisphosphonates: −0.058 (0.258); teriparatide: 0.021 (0.044), *p* = 0.191).

There was no significant difference in the change in HbA1c among groups (no medication, 0.273 (0.775)%; bisphosphonates, 0.326 (0.845)%; and teriparatide, 0.033 (0.407)%; *p* = 0.401).

### 3.6. Data Analyses after Excluding Patients Who Could Not Take Medications for One Year

Table 3 shows the values of BMD and TBS and their changes among the participants who could use the medications for one year (25 of 28 patients in the bisphosphonates group and 10 of 21 patients in the teriparatide group). The results were similar to those of the entire study population. However, the change in BMD of the femoral neck in the teriparatide group (0.027 (0.016) g/cm2, 5.113(3.115)%) was significantly increased compared to the other groups (no medication group: −0.002 (0.034) g/cm2, *p* = 0.016, −0.114(5.315)%, *p* = 0.009; bisphosphonates group: −0.0006 (0.025) g/cm2, *p* = 0.043, 0.142(4.833)%, *p* = 0.025).

### 3.7. Safety Evaluation

Among the 28 patients who received bisphosphonates, one patient stopped due to patient preference. Among the 21 patients who received weekly teriparatide, seven patients stopped due to fever (1 patient), skin rash (1 patient), loss of appetite (1 patient), and high costs of treatment (4 patients).

## 4. Discussion

This study investigated the effect of weekly teriparatide on BMD in osteoporosis patients with T2DM. Teriparatide resulted in an improvement in BMD of the lumbar spine, whole femur, and femoral neck. BMD of the femoral neck tended to improve more in the teriparatide group than in the other groups, although there was no statistically significant difference. Further, teriparatide significantly improved BMD of the femoral neck among the participants who could use medications for one year. Furthermore, although there was no statistically significant difference, TBS in the teriparatide group showed an increasing trend compared to the other groups.

The goal of diabetes treatment is to prevent the onset of complications and to improve patients’ quality of life. However, multiple meta-analyses have shown that the risk of fracture is higher in patients with diabetes, with a 1.32-fold higher risk of total fracture. Further, the risk of fracture due to diabetes is considered to be site-specific, and in particular, the risk of proximal femur fracture was reported to be 1.77 times higher [6]. Osteoporosis and its associated fracture risk is one of the complications of diabetes and is a factor that impairs healthy life expectancy, especially in older patients with T2DM. 

In this study, weekly teriparatide significantly increased the BMD of the femoral neck in patients with T2DM. Specifically, the change in BMD of the femoral neck in the teriparatide group (0.027 (0.016) g/cm^2^, 5.113 (3.115)%) was significantly increased compared to the other groups (no medication group: −0.002 (0.034) g/cm^2^, *p* = 0.016, −0.114 (5.315)%, *p* = 0.009; bisphosphonates group: −0.0006 (0.025) g/cm^2^, *p* = 0.043, 0.142 (4.833)%, *p* = 0.025). A previous study revealed that the degree of increase in BMD with daily injections of teriparatide was greater than that of alendronate (11% (5) vs. 4% (4), *p* < 0.001) [26]. This is similar to our results. Another study showed that greater improvements in BMD were strongly associated with greater reductions in vertebral and hip fractures. Therefore, we believe that the increase in BMD in this study will lead to a decrease in fractures. Furthermore, the percent change of BMD was −0.114% (5.315) with no medicine, 0.142% (4.833) with bisphosphonates, and 5.113% (3.115) with teriparatide. A previous study showed that we might expect a 16% or 40% reduction in hip fracture risk for a 2% or 6% improvement in total hip BMD [27]. Therefore, we believe that the changes in this study, especially the 5.113 (3.115)% change with teriparatide, may be effective in reducing fractures. Another study reported that compared to people without diabetes, T2DM patients treated with daily teriparatide resulted in a greater increase in the femoral neck BMD, and usage of daily teriparatide significantly reduced the incidence of new non-vertebral fractures [21]. Another study with an indirect comparison reported that teriparatide was more effective than bisphosphonates in significantly reducing vertebral fractures [28]. As mentioned above, osteoporosis due to diabetes causes fractures that contribute to bone deterioration, especially in the femoral neck, where cortical bone predominates. Moreover, the risk of fractures in osteoporosis associated with diabetes is higher than that predicted by BMD, suggesting that worsening bone quality contributes to diabetes-induced bone fragility. In fact, bone strength as measured by micro indentation is reduced in patients with T2DM [29]. Mechanisms of bone quality deterioration include a general decrease in bone remodeling, cortical bone fragility due to increased cortical bone porosity, trabecular bone fragility due to changes in trabecular bone microstructure, and decreased strength of type 1 collagen fibers due to increased AGE cross-linking. Teriparatide has been reported to improve bone quality by inhibiting non-physiological cross-linking, such as pentosidine cross-linking caused by AGEs, and promoting physiological cross-linking by production of lysyl oxidase in osteoblasts, which may be effective as a treatment for osteoporosis in patients with T2DM [16]. TBS, which reflects bone quality, in the teriparatide group was increased compared to the other groups, although the difference was not significant. These results suggest that teriparatide improves bone quality in patients with T2DM. Furthermore, these results are consistent with the results of a post hoc analysis of the Abaloparatide Comparator Trial In Vertebral Endpoints (ACTIVE), which demonstrated the efficacy of daily teriparatide administration on trabecular bone score in T2DM patients with osteoporosis [30]. In the Japanese Osteoporosis INtervention Trial-05 (JOINT05), teriparatide significantly reduced the incidence of morphological vertebral fractures compared with alendronate in women with primary osteoporosis at high risk of fracture [31]. This study reported comparable effects between the two groups for BMD-lumbar spine, whole femur, femoral neck, and forearm. These are different from our study, but we consider that the small number of study subjects may have influenced the results. 

Previous studies have reported that bisphosphonates target osteoclast, osteoblast, and osteocyte activity and improve BMD, while teriparatide enhances osteoblast function, induces new bone matrix, and increases BMD. In our study, teriparatide improved BMD in the lumbar spine, whole femur, and femoral neck, while bisphosphonates did not clearly increase BMD. On the other hand, side effects include osteonecrosis of the jaw for bisphosphonates [13], and nausea, arthralgia, hypertension, headache, and hypercalcemia have been reported for teriparatide [18].

Despite its advantages, the safety and tolerability of teriparatide must be considered. In our study, teriparatide was effective in patients who could continue treatment; however, 7 out of 21 patients dropped out, which may suggest that under real circumstances, many patients have difficulty in continuing treatment for a long period of time due to price and side effects.

In addition to financial reasons, some patients dropped out due to symptoms such as fever, skin eruption, and loss of appetite. Consistent with our own observations, an interview form has shown that the frequency of fever, skin eruption, and loss of appetite are all less than 0.1 to 5% [32]. 

This study has several limitations that should be noted. First, this was a retrospective cohort study; therefore, some bias may be present. Further studies, especially randomized controlled trials, are needed to compare the effects of weekly teriparatide and bisphosphonates on diabetic-complicated osteoporosis. Second, since the sample size is not adequate to overcome variations, large-scale additional studies are needed. Third, events such as fractures were not evaluated. Therefore, further studies are warranted to further our understanding of the risk of fractures in diabetic patients. Fourth, to clarify the usefulness of teriparatide, calculation of the sample size is desirable. However, this was a retrospective cohort study, and thus we did not calculate the sample size. Fifth, according to previous reports, antidiabetic medications, especially insulin and thiazolidinediones, are fracture risks [33,34]. However, due to the small number of patients in this study, the effects of these medications were not examined. Further studies are needed to consider the correlation between antidiabetic medication classes and fracture risk.

## 5. Conclusions

In conclusion, this study showed that teriparatide significantly increased BMD at the femoral neck in patients with osteoporosis complicated by T2DM compared to bisphosphonates and no medication. Further precise analysis, such as randomized controlled trials, would be necessary to clarify this causal relationship.

## Figures and Tables

**Figure 1 medicina-58-00481-f001:**
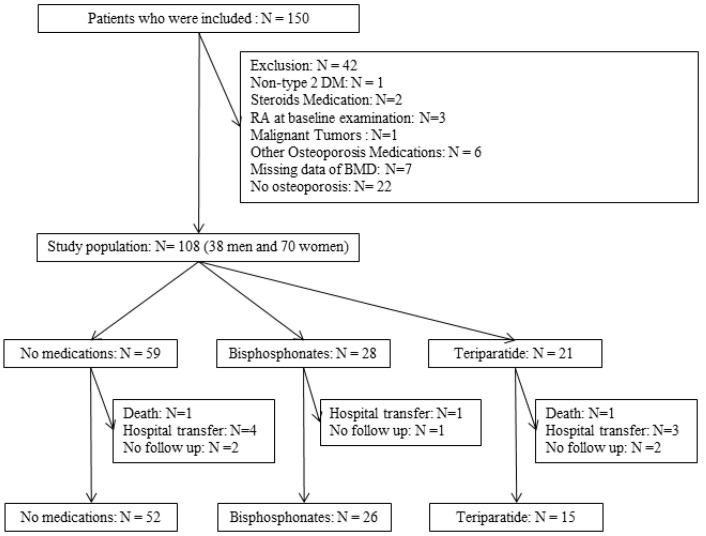
Inclusion and exclusion flow.

**Table 1 medicina-58-00481-t001:** Characteristics of study participants at the baseline examination.

	All	No Medication	Bisphosphonate Usage	Teriparatide Usage	*p*
*n*	93	52	26	15	―
Age (years)	112.0 (93.5–122.5)	115.5 (104.3–129.8) †	94.5 (79.8–112.5)	120.0 (88.0–123.0)	0.001
Men (%)	34.4	44.2	23.1	20	0.079
Height	156.0 (150.5–162.5)	157.5 (151.6–166.4)	152.2 (149.0–159.0)	153.0 (150.0–162.0)	0.108
Body weight	58.0 (51.5–67.0)	59.5 (55.1–67.0)	54.0 (48.0–61.3)	55.0 (48.0–68.0)	0.067
Body mass index (kg/m^2^)	24.2 ± 3.7	24.5 ± 3.3	23.9 ± 4.5	23.7 ± 3.7	0.703
Biguanide (yes, %)	38.7	46.2	15.4	53.3	0.014
Thiazolidinediones (yes, %)	3.2	3.9	3.8	0	0.742
Sulfonylurea (yes, %)	18.3	17.3	30.8	0	0.047
Glinide (yes, %)	7.5	9.6	7.7	0	0.461
DPP4 inhibitor (yes, %)	45.2	38.5	57.7	46.7	0.272
SGLT2 inhibitor (yes, %)	15.1	13.5	7.7	33.3	0.077
α glucosidase inhibitor (yes, %)	16.1	17.3	19.2	6.7	0.540
GLP-1 receptor agonist (yes, %)	30.1	36.5	23.1	20	0.307
Insulin (yes, %)	25.8	28.9	30.8	6.7	0.178
Exercise (yes, %)	43	40.4	50	40	0.698
Smoking states					0.450
Never-smoker (%)	67.7	63.5	80.8	60	
Ex-smoker (%)	20.4	25.0	11.5	20	
Current smoker (%)	11.8	11.5	7.7	20	
Neuropathy (yes, %)	15.1	17.3	15.4	6.7	0.596
Nephropathy (yes, %)	41.9	46.2	38.5	33.3	0.617
Retinopathy (yes, %)	20.4	23.1	19.2	13.3	0.701
HbA1c (%)	7.20 (6.80–7.80)	7.30 (6.80–7.75)	7.25 (6.80–7.95)	7.00 (6.40–7.60)	0.450
Duration of diabetes	12.00 (5.00–18.50)	13.00 (5.25–18.75)	13.50 (7.25–20.50)	8.00 (4.00–13.00)	0.233
Trabecular bone score (*n* = 92)	1.32 ± 0.10	1.35 ± 0.10 ‡	1.30± 0.09	1.28 ± 0.08	0.011
Bone mineral density of the lumbar spine	0.9 ± 0.2	1.0 ± 0.2 †‡	0.8 ± 0.2	0.9 ± 0.2	<0.001
Bone mineral density of the whole femur	0.70 (0.66–0.87)	0.83 (0.72–0.92) †‡	0.68 (0.62–0.77)	0.72 (0.62–0.79)	<0.0001
Bone mineral density of the femoral neck	0.61 (0.54–0.72)	0.69 (0.56–0.76) †‡	0.54 (0.47–0.60)	0.56 (0.50–0.61)	<0.0001

Data are percent of subjects or mean ± SD. *p* values by one-way analysis of variance for continuous variables or Kruskal–Wallis test and chi-squared test for categorical variables. The analyses of continuous variables among the three groups were performed by Tukey HSD test or steel Dwass test: †, *p* < 0.05 versus bisphosphonate, ‡, *p* < 0.05 versus teriparatide. DPP4, Dipeptidyl Peptidase-4; SGLT2, Sodium-glucose transporter 2; GLP-1, Glucagon-like peptide-1.

**Table 2 medicina-58-00481-t002:** The change of bone mineral density or trabecular bone score.

	No Medication				Bisphosphonate Usage				Teriparatide Usage				*p*
	Before	After	BMD Change	BMD% Change	Before	After	BMD Change	BMD% Change	Before	After	BMD Change	BMD% Change	
Bone mineral density of the lumbar spine	1.02 (0.19)	1.04 # (0.19)	0.011 † (0.031)	1.102 (3.155) †‡	0.83 (0.24)	0.87 # (0.25)	0.044 (0.044)	5.519 (5.631)	0.86 (0.18)	0.89 # (0.16)	0.029 (0.030)	4.080 (4.747)	0.001 0.001
Bone mineral density of the whole femur	0.83 (0.15)	0.83 (0.16)	−0.003 (0.055)	−0.308 (6.655)	0.70 (0.11)	0.69 (0.11)	−0.007 (0.038)	−0.912 (5.769)	0.73 (0.15)	0.74 # (0.15)	0.014 (0.019)	1.951 (2.944)	0.351 0.319
Bone mineral density of the femoral neck	0.68 (0.15)	0.68 (0.14)	−0.002 (0.034)	−0.114 (5.315)	0.54 (0.10)	0.54 (0.10)	−0.0001 (0.024)	−0.216 (4.751)	0.59 (0.16)	0.60 # (0.16)	0.017 (0.022)	3.230 (4.221)	0.091 0.075
Trabecular bone score (*n* = 92)	1.34 (0.10)	1.34 (0.09)	−0.007 (0.051)		1.30 (0.09)	1.24 (0.27)	−0.058 (0.258)		1.28 (0.08)	1.30 (0.07)	0.021 (0.044)		0.191

Data are mean ± SD. *p* values determined by one-way analysis of variance for continuous variables and chi-squared test for categorical variables. For *p* value, the upper represents the difference of BMD change, and the lower represents the difference of BMD% change. The differences among three groups were determined by the Tukey HSD test: †, *p* < 0.05 versus bisphosphonate; ‡, *p* < 0.05 versus teriparatide. The differences of BMD and TBS between before and after were determined by the paired *t*-test: #, *p* < 0.05.

**Table 3 medicina-58-00481-t003:** The change of bone mineral density or trabecular bone score among the patients who continued to the medication.

	No Medication				Bisphosphonate Usage				Teriparatide Usage				*p*
	Before	After	BMD Change	BMD% Change	Before	After	BMD Change	BMD% Change	Before	After	BMD Change	BMD% Change	
Bone mineral density of the lumbar spine	1.02 (0.19)	1.04 # (0.19)	0.011 † (0.031)	1.102 (3.155) †‡	0.80 (0.20)	0.84 # (0.21)	0.043 (0.045)	5.573 (5.740)	0.80 (0.17)	0.84 # (0.16)	0.036 (0.033)	5.336 (5.360)	0.001 <0.0001
Bone mineral density of the whole femur	0.83 (0.15)	0.83 (0.16)	−0.003 (0.055)	−0.308 (6.655)	0.69 (0.11)	0.69 (0.11)	−0.007 (0.039)	−0.964 (5.881)	0.70 (0.12)	0.72 # (0.13)	0.016 (0.019)	2.261 (3.064)	0.420 0.373
Bone mineral density of the femoral neck	0.68 (0.15)	0.68 (0.14)	−0.002 ‡ (0.034)	−0.114 (5.315) ‡	0.54 (0.10)	0.54 (0.10)	−0.0006 (0.025) ‡	0.142 (4.833) ‡	0.55 (0.11)	0.57 # (0.11)	0.027 (0.016)	5.113 (3.115)	0.020 0.011
Trabecular bone score (*n* = 87)	1.35 (0.10)	1.34 (0.09)	−0.007 (0.051)		1.29 (0.09)	1.23 (0.27)	−0.060 (0.264)		1.26 (0.08)	1.29 (0.07)	0.027 (0.051)		0.205

*p* values determined by one-way analysis of variance for continuous variables and chi-squared test for categorical variables. For *p* value, the upper represents the difference of BMD change, and the lower represents the difference of BMD% change. The differences among three groups were determined by the Tukey HSD test: †, *p* < 0.05 versus bisphosphonate; ‡, *p* < 0.05 versus teriparatide. The differences of BMD and TBS between before and after were determined by the paired *t*-test: #, *p* < 0.05.

## Data Availability

The data that support the findings of this study are available from the corresponding author, Y.H., upon reasonable request.

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
