# Peer review of "Effect of Teriparatide on Bone Mineral Density and Trabecular Bone Score in Type 2 Diabetic Patients with Osteoporosis: A Retrospective Cohort Study"

_medicina, 2022, doi:10.3390/medicina58040481_

Round 1

Reviewer 1 Report

This is an observational retrospective study with a relatively small population that analysed the differences in BMD and TBS in patients with type 2 diabetes and osteoporosis. The authors used only univariate analysis methods.

Statistical analysis

Page 3, Line 108. Do the variables have a normal distribution? Given the small number of patients in some groups, median and interquartile range could be used if the variables have skewed distribution.

Results

Page 3, Line 132. After the second paragraph where the authors explained the follow-up, I suggest that the final number of patients included for analysis should be specified (the one from the Abstract).  

Data in Table 1 are difficult to read. May the authors could keep only the percentage for “yes” for the categorical variables.

Page 6, Line 168. The authors already mentioned Table 2. This sentence could be deleted.

Page 7, Line 188. The results in Table 2 are before and after receiving treatment for 92 subjects. In Table 1 there are 93 subjects. Please explain the difference in the text.

Also, in the text “10 of 21 patients in the teriparatide group used the medication for one year ”, so the total number for initial and final comparison cannot be the same (92), because at the enrollment there  were 15 patients.

Other studies reported the results as % of improvement in BMD, this would be clearer for the reader to use the information in clinical practice. Consider calculating the percentages. In the <Discussion> section I would comment, for example, if a 0.01 g/cm2 increase in BMD of the whole femur is clinically significant compared with other studies.

Page 8 Line 228. Add the actual numbers (%, g/cm2) for the increase in BMD - reference [23] and compare it with your results.

Page 8 Line 244. Explain abbreviation “ACTIVE” and “JOINT05”.

Page 8 Line 265. Have you calculated the sample size needed for the size effect that you assumed for teriparatide at the beginning of the study (“since the sample size is not so big  to overcome variations”)?

Have you analysed the correlation between antidiabetic medication classes and fracture risk? I consider this important since previous RCT and meta-analyses showed an increased risk with thiazolidindiones (https://www.frontiersin.org/articles/10.3389/fendo.2013.00011/full); or insulin in real-world registries (https://link.springer.com/article/10.1007/s00198-018-4581-y).

Author Response

Page 3, Line 108. Do the variables have a normal distribution? Given the small number of patients in some groups, median and interquartile range could be used if the variables have skewed distribution.

Response

Thank you for your suggestion. As you say, to use the median or interquartile range is desirable for the variables with skewed distributions. According to your suggestion, we have reanalyzed the variables and changed the expression of some of variables in the Statistical Analysis section, described as below (Page 3, Line 116) and Table 1.

Statistical Analysis

“The normal distributions were evaluated by Shapiro-Wilk test. We show the data as means (SD), median (1st quartile -3rd quartile) or frequencies of potential confounding variables. Patients were divided into the following three groups: 1) no medication, 2) bisphosphonates (alendronic acid or risedronate), and 3) teriparatide (Teribone™ 56.5 µg via subcutaneous injection once a week). Then, the differences in the continuous variables and categorical variables were evaluated by one-way ANOVA and Tukey-Kramer test or Kruskal-Wallis test and steel dwass test, and the chi-square test, respectively.”

Page 3, Line 132. After the second paragraph where the authors explained the follow-up, I suggest that the final number of patients included for analysis should be specified (the one from the Abstract).

Response

Thank you for your suggestion. According to your suggestion, we have added the final number of patients included for analysis in the Results section, described as below (Page 3, Line 130).

Results

“Finally, 93 patients, of whom 52 received no medication, 26 received bisphosphonates, and 15 received weekly teriparatide, were included in the analysis.”

Data in Table 1 are difficult to read. May the authors could keep only the percentage for “yes” for the categorical variables.

Response

Thank you for your comment. According to your comment, we have revised the Table 1.

Page 6, Line 168. The authors already mentioned Table 2. This sentence could be deleted.

Response

Thank you for your comment. According to your comment, we have revised it.

Page 7, Line 188. The results in Table 2 are before and after receiving treatment for 92 subjects. In Table 1 there are 93 subjects. Please explain the difference in the text.

Response

We are sorry for confusing with you. We included 93 patients in this study. On the other hand, the TBS data was available for 92 patients, so 92 patients were used for the analysis of TBS. We have added this point in the Results section described as below (Page 6, Line 154).

Results

“Although 93 patients were included in this study, the TBS data was available for 92 patients, so 92 patients were used for the analysis of TBS.”

Also, in the text “10 of 21 patients in the teriparatide group used the medication for one year ”, so the total number for initial and final comparison cannot be the same (92), because at the enrollment there were 15 patients.

Response

We are sorry for confusing with you. Our study included 108 patients, of whom 59 received no medication, 28 received bisphosphonates, and 21 received weekly teriparatide. Among them, 93 patients, of whom 52 received no medication, 26 received bisphosphonates, and 15 received weekly teriparatide, received follow-up examination. Thus, 93 patients were included in the analysis. As for teriparatide usage, teriparatide was started in 21 patients, but as shown in Fig. 1, one patient died, 3 patients were transferred to another hospitals, 2 patients did not receive follow-up, and 15 patients could be followed up after one year, of whom 10 patients were able to continue for one year. Therefore, among the 21 patients, 10 patients were able to continue for one year. According to your suggestion, we have revised the Results section described as below (Page 3, Line 130).

Results

“Finally, 93 patients, of whom 52 received no medication, 26 received bisphosphonates, and 15 received weekly teriparatide, were included in the analysis. In addition, among 28 patients who received bisphosphonates, 25 patients were able to continue for one year, and among the 21 patients who received weekly teriparatide, 10 patients were able to continue for one year.”

Other studies reported the results as % of improvement in BMD, this would be clearer for the reader to use the information in clinical practice. Consider calculating the percentages. In the <Discussion> section I would comment, for example, if a 0.01 g/cm2 increase in BMD of the whole femur is clinically significant compared with other studies.

Response

Thank you for your valuable suggestion. Certainly, it is difficult to understand whether a 0.01 g/cm2 increase in BMD is clinically significant. According to your suggestion, we have investigated the percent change of BMD in all groups, and it was -0.114 [5.315] % with no medicine, 0.142 [4.833] % with bisphosphonates, and 5.113 [3.115] % with teriparatide. In a previous study, it was reported that we might expect a 16% or 40% reduction in hip fracture risk for a 2% or 6% improvement in total hip BMD. Therefore, we believe that the changes in this study, especially the 5.113 [3.115] % change with teriparatide, may be effective in reducing fractures. We summarize this point in the Discussion section as follows (Page 8, Line 238).

Discussion

“The percent change of BMD was -0.114 [5.315] % with no medicine, 0.142 [4.833] % with bisphosphonates, and 5.113 [3.115] % with teriparatide. A previous study show that we might expect a 16% or 40% reduction in hip fracture risk for a 2% or 6% improvement in total hip BMD [26]. Therefore, we believe that the changes in this study, especially the 5.113 [3.115] % change with teriparatide, may be effective in reducing fractures.”

Reference

  1. Mary L Bouxsein, Richard Eastell, Li-Yung Lui, Lucy A Wu, Anne E de Papp, et al. (2019) Change in Bone Density and Reduction in Fracture Risk: A Meta-Regression of Published Trials. Journal of Bone and Mineral Research 34: 632-642. https://doi.org/10.1002/jbmr.3641.

Page 8 Line 228. Add the actual numbers (%, g/cm2) for the increase in BMD - reference [23] and compare it with your results.

Response

Thank you for your valuable suggestion. According to your suggestion, we have added the actual numbers (%, g/cm2) for the increase in BMD of the previous study and compared the results in the Discussion section descried as below (Page 8, Line 228).

Discussion

“In this study, weekly teriparatide significantly increased the BMD of the femoral neck in patients with T2DM. Specifically, the change in BMD of the femoral neck in the teriparatide group (0.027 [0.016] g/cm2, 5.113 [3.115] %) was significantly increased compared to the other groups (no medication group, −0.002 [0.034] g/cm2, p = 0.016, -0.114 [5.315] % , p = 0.009; and bisphosphonates group, −0.0006 [0.025] g/cm2, p = 0.043, 0.142 [4.833] %, p = 0.025). A previous study revealed that the degree of increase in BMD with daily injections of teriparatide was greater than that of alendronate (11 [5] vs. 4 [4] %, P < 0.001) [25]. This is similar to our results.”

Page 8 Line 244. Explain abbreviation “ACTIVE” and “JOINT05”.

Response

Thank you for your comment. According to your comment, we have added the description in the Discussion section (Page 9, Line 262).

Discussion

“Furthermore, these results are consistent with the results of a post hoc analysis of the Abaloparatide Comparator Trial In Vertebral Endpoints (ACTIVE) trial, which demonstrated the efficacy of daily teriparatide administration on trabecular bone score in type 2 diabetic patients with osteoporosis [29]. In the Japanese Osteoporosis INtervention Trial-05 (JOINT05) trial, teriparatide significantly reduced the incidence of morphological vertebral fractures compared with alendronate in women with primary osteoporosis at high risk of fracture [30].”

Page 8 Line 265. Have you calculated the sample size needed for the size effect that you assumed for teriparatide at the beginning of the study (“since the sample size is not so big to overcome variations”)?

Response

As you indicated, the calculation of sample size is important, but this study is a retrospective cohort study, so no calculations were performed. Thus, we have mentioned this point as one of the limitations of this study in the Discussion section as below. (Page 9, Line 295).

Discussion

“Fourth, to clarify the usefulness of teriparatide, calculation of the sample size was desirable. However, this was a retrospective cohort study, we did not calculate the sample size.”

Have you analysed the correlation between antidiabetic medication classes and fracture risk? I consider this important since previous RCT and meta-analyses showed an increased risk with thiazolidindiones (https://www.frontiersin.org/articles/10.3389/fendo.2013.00011/full); or insulin in real-world registries (https://link.springer.com/article/10.1007/s00198-018-4581-y).

Response

Thank you for your valuable suggestion. As you say, the use of antidiabetic medications, especially insulin and TZDs, is a fracture risk, and we wanted to examine the impact of these medications. Unfraternally, however, due to the small number of patients, we did not perform the analysis. According to your suggestion, we have added this point in the Limitation in the Discussion section as follows (Page 9, Line 297).

Discussion

“Fifth, according to previous reports, antidiabetic medications, especially insulin and thiazolidinediones, are one of the fracture risks [32, 33]. However, due to the small number of patients in this study, the effects of these medications were not examined. Further studies are needed to consider the correlation between antidiabetic medication classes and fracture risk.”

Reference

  1. Bazelier MT, de Vries F, Vestergaard P, Herings RM, Gallagher AM, et al. (2013) Risk of fracture with thiazolidinediones: an individual patient data meta-analysis. Front Endocrinol (Lausanne) 26:11. https://doi.org/10.3389/fendo.2013.00011.
  2. Losada E, Soldevila B, Ali MS, Martínez-Laguna D, Nogués X, et al. (2018) Real-world antidiabetic drug use and fracture risk in 12,277 patients with type 2 diabetes mellitus: a nested case-control study. Osteoporosis Int. 29: 2079-2086. https://doi.org/10.1007/s00198-018-4581-y.

Reviewer 2 Report

The manuscript entitled “Effect of teriparatide on bone mineral density and trabecular bone score in type 2 diabetic patients with osteoporosis; a retro-spective cohort study” submitted to Medicina aims investigate the effect of teriparatide on BMD in patients with T2DM compared with bisphosphonates or no medication.

The manuscript appears interesting however, many manuscripts on osteoporosis treatment strategies are documented in the literature.

I have some suggestions to improve deeply the quality of the manuscript, enriching the text with further notions.

English form: Minor revision

Introduction: I suggest to improve the descriptive part about BPs and teriparatide. In particular, I suggest to describe the possible adverse effects of these drug use (MRONJ, hypercalcemia and urinary calcium excretion ) referring to these recent studies [PMID: 32615096 - PMID: 33086890 -  PMID: 15262455 - PMID: 26902094]

Materials and methods: Authors structured in the right way this section.

Results: Sample features are well described in this section; statistical analysis showed interesting results enriching manuscript quality.

Discussion: Discussion are complete, including in a detailed part the study limitations. I suggest to add a part about the systemic effect of these drug use.

Conclusion: This part is in accordance to the hypostasis and study results.

After the suggested improvements, this study must to be re-evaluate.

Author Response

Introduction: I suggest to improve the descriptive part about BPs and teriparatide. In particular, I suggest to describe the possible adverse effects of these drug use (MRONJ, hypercalcemia and urinary calcium excretion) referring to these recent studies [PMID: 32615096 - PMID: 33086890 -  PMID: 15262455 - PMID: 26902094].

Response

Thank you for your comment. According to your comment, we have added the description about the possible adverse effects of these medication use in the Introduction section descried as below (Page 2, Line 50).

Introduction

“Despite bisphosphonates' well-known therapeutic potential, they also displayed important side effects, among which is bisphosphonates-related osteonecrosis of the jaw, by targeting osteoclast activities, osteoblast, and osteocyte behavior [13].”

“On the other hand, the most frequently reported adverse events were nausea (12.5 %), arthralgia (11.7 %), hypertension (8.9 %), and headache (6.9 %). Hypercalcemia was reported in 5 % of the patients [17].”

References

  1. Anna Di Vito, E Chiarella, F Baudi, P Scardamaglia, A Antonelli, et al. (2020) Dose-Dependent Effects of Zoledronic Acid on Human Periodontal Ligament Stem Cells: An In Vitro Pilot Study. Cell transplantation 29: 1555-3892. https://doi.org/10.1177/0963689720948497.
  2. R. Lindsay, J. H. Krege, F. Marin, L. Jin, J. J. Stepan. (2016) Teriparatide for osteoporosis: importance of the full course. Osteoporos Int 27: 2395–2410. https://doi.org/ 10.1007/s00198-016-3534-6.

Materials and methods: Authors structured in the right way this section.

Response

Thank you for your comment.

Results: Sample features are well described in this section; statistical analysis showed interesting results enriching manuscript quality.

Response

Thank you for your comment.

Discussion: Discussion are complete, including in a detailed part the study limitations. I suggest to add a part about the systemic effect of these drug use.

Response

Thank you for your suggestion. According to your suggestion, we have added a part about the systemic effect of these drug use in the Discussion descried as below (Page 9, Line 273).

Discussion

“Previous studies have reported that bisphosphonates target osteoclast, osteoblast, and osteocyte activity and improve BMD, while teriparatide enhances osteoblast function, induces new bone matrix, and increases BMD. In our study, teriparatide improved BMD in the lumbar spine, whole femur, and femoral neck, while bisphosphonates did not clearly increase BMD. On the other hand, side effects include osteonecrosis of the jaw for bisphosphonates [13] and nausea, arthralgia, hypertension, headache, and hypercalcemia have been reported for teriparatide [17].”

References

  1. Anna Di Vito, E Chiarella, F Baudi, P Scardamaglia, A Antonelli, et al. (2020) Dose-Dependent Effects of Zoledronic Acid on Human Periodontal Ligament Stem Cells: An In Vitro Pilot Study. Cell transplantation 29: 1555-3892. https://doi.org/10.1177/0963689720948497.
  2. R. Lindsay, J. H. Krege, F. Marin, L. Jin, J. J. Stepan. (2016) Teriparatide for osteoporosis: importance of the full course. Osteoporos Int 27: 2395–2410. https://doi.org/ 10.1007/s00198-016-3534-6.

Conclusion: This part is in accordance to the hypostasis and study results.

Response

Thank you for your comment.

Round 2

Reviewer 1 Report

You have improved the clarity of your results and the quality of discussions. Table 2 and table 3 remain difficult to read. Consider removing the column with "paired t test" and just flag the significant differences (with mentioning this in the legend).

Author Response

Response to Reviewer 1

You have improved the clarity of your results and the quality of discussions. Table 2 and table 3 remain difficult to read. Consider removing the column with "paired t test" and just flag the significant differences (with mentioning this in the legend).

Response

Thank you for your suggestion. As you say, we have revised the Table 2 and Table 3.

Reviewer 2 Report

the authors have made some changes following the suggestions of the reviewers.

I suggest to implement the part of adverse events by also listing the clinical counterpart that these drugs can cause, not only from a molecular point of view [PMID: 32615096 - PMID: 15262455].
This may help a wide range of readers to understand both molecularly and clinically the effects of these drugs.

The manuscript needs further revision after the suggested refinements.

Author Response

Response to Reviewer 2

I suggest to implement the part of adverse events by also listing the clinical counterpart that these drugs can cause, not only from a molecular point of view [PMID: 32615096 - PMID: 15262455]. This may help a wide range of readers to understand both molecularly and clinically the effects of these drugs.

Response

Thank you for your comment. According to your comment, we have added the description about the clinical counterpart that these drugs can cause, not only from a molecular point of view in the Introduction section descried as below (Page 2, Line 53, Line 64).

Introduction

“Recently, minimally invasive surgical treatment from early stage on the Medication-related osteonecrosis of the jaw has been recommended to prevent the evolution to more advanced stages and to promote downstaging of the lesion [14].”

“The calcium concentration increases transiently, beginning 2 hours after dosing, reaching a maximum concentration between 4 and 6 hours (median increase, 0.4 mg/dL), decreasing 6 hours after dosing, and returning to baseline values by 16 to 24 hours after dosing. Persistent hypercalcemia was not observed in clinical trials of teriparatide. If persistent hypercalcemia is detected, teriparatide should be discontinued and the patient should be assessed for the cause of the hypercalcemia [19].”

References

14. Amerigo Giudice, Selene Barone, Federica Diodati, Alessandro Antonelli, Riccardo Nocini, et al. (2020) Can Surgical Management Improve Resolution of Medication-Related Osteonecrosis of the Jaw at Early Stages? A Prospective Cohort Study. Journal of Oral and Maxillofacial Surgery 78: 1986-1999. https://doi.org/10.1016/j.joms.2020.05.037.

19. Elaena Quattrocchi, Helen Kourlas. (2004) Teriparatide: A review. Clinical Therapeutics 26: 841-854. https://doi.org/10.1016/S0149-2918(04)90128-2.